# Using A Low-Cost Sensor Array and Machine Learning Techniques to Detect Complex Pollutant Mixtures and Identify Likely Sources

**DOI:** 10.3390/s19173723

**Published:** 2019-08-28

**Authors:** Jacob Thorson, Ashley Collier-Oxandale, Michael Hannigan

**Affiliations:** 1Mechanical Engineering, University of Colorado Boulder, Boulder, CO 80309, USA; 2Environmental Engineering, University of Colorado, Boulder, Boulder, CO 80309, USA

**Keywords:** VOCs, emissions, low-cost sensors, sensor arrays, classification, regression

## Abstract

An array of low-cost sensors was assembled and tested in a chamber environment wherein several pollutant mixtures were generated. The four classes of sources that were simulated were mobile emissions, biomass burning, natural gas emissions, and gasoline vapors. A two-step regression and classification method was developed and applied to the sensor data from this array. We first applied regression models to estimate the concentrations of several compounds and then classification models trained to use those estimates to identify the presence of each of those sources. The regression models that were used included forms of multiple linear regression, random forests, Gaussian process regression, and neural networks. The regression models with human-interpretable outputs were investigated to understand the utility of each sensor signal. The classification models that were trained included logistic regression, random forests, support vector machines, and neural networks. The best combination of models was determined by maximizing the F_1_ score on ten-fold cross-validation data. The highest F_1_ score, as calculated on testing data, was 0.72 and was produced by the combination of a multiple linear regression model utilizing the full array of sensors and a random forest classification model.

## 1. Introduction

Understanding the causes of degraded air quality at a local scale is a challenging but important task. Making the task especially difficult is the complexity of possible sources of pollutants, each of which produce a wide variety of chemical species. These pollutant mixtures often include a component of volatile organic compounds (VOCs), which are important because of both direct toxicological health impacts as well as the impacts caused by secondary products. Examples of mechanisms that result in secondary products from VOC emission include condensation into particulate matter (PM) and reaction with other compounds that result in increased tropospheric ozone. There is also no single instrument that can fully quantify the complex and dynamic changes of ambient air composition needed to identify local sources, and communities directly affected by air quality problems often do not have the resources to mount an extensive measurement campaign needed to identify likely sources. Quantifying the impact of and identifying these sources at a higher resolution than is currently possible would contribute to regulatory, scientific, and public health goals.

Regulators could benefit from this type of information by both understanding what types of sources to target in order to address specific problems and by understanding what industries are likely to be affected by regulations targeting specific compounds. For example, a critical piece of legislation that governs toxic air pollutants is the list of Hazardous Air Pollutants (HAPs), which was produced and has been updated by the US Environmental Protection Agency (EPA) [1]. When determining whether a specific HAP is likely to affect a community, it is important to understand the nature of sources on a broad range of scales because some compounds only affect communities near to their source, while others can have global effects. California’s recent state bill, AB 617, is an example of the regulatory application of a low-cost measurement approach to understanding VOC sources. The bill is intended to address local air quality problems by both supporting community-based monitoring of air toxics and using the data gathered by those monitors to address important sources of hazardous compounds [2].

From a scientific perspective, the location and nature of emission sources can provide important information for atmospheric chemistry models. These models are used to understand the production of important secondary products including forms of particulate matter (PM), ozone, and other compounds with health and environmental impacts. Currently, these models use some combination of emission factors, measurement campaigns, and back-propagation modeling to estimate emissions from important sources within urban areas. Although continuous measurements that are used as inputs to these models are often spaced on the order of tens to hundreds of kilometers apart, it has been shown that the concentration of chemically important compounds vary on the scale of single to tens of meters in urban environments [3,4]. Dense, continuous measurements of air quality within cities could provide insights into small-scale variations on a real-time basis that are difficult to capture using current techniques. For example, several studies have demonstrated the ability of dense networks or mobile platforms with low-cost sensors to quantify the variation of pollutants like NO_2_, CO, O_3_, and PM within urban locales [5,6,7,8].

From a public health perspective, networks of low-cost sensors may provide the ability to understand real-life exposures for individuals to categories of compounds with suspected health impacts. The first examples of these studies using low-cost sensors have included some combination of small, portable measurement devices or dense networks and GPS locations of the individual. Once continuous and highly specific exposure data becomes available, it may be possible to couple that data with health outcomes to better understand the relation between exposures and outcomes at typical, ambient exposures (as opposed to workplace data, where exposure levels are typically much higher). Developing affordable methods to study air quality at local scales is important as it may help identify and address environmental justice problems that may contribute to higher disease burden in vulnerable communities [9].

The work presented here provides a proof-of-concept for low-cost analytical tools to accomplish this goal on a more accessible scale. In our proposed methodology, analytical methods were applied to low-cost sensors in order to estimate the concentrations of important compounds and to predict the “class” of source that is emitting those compounds. Using this methodology, community members could have a “first pass” estimate of what they are being exposed to and what the likely source(s) of those compounds are. With these more human-readable outputs, community groups would be better able to make decisions such as when and where to take air samples, what pollutant species to focus on, or what part of their local government to reach out to for a follow-up measurement campaign with possible regulatory implications.

Previous studies have addressed components of the work entailed here but have not addressed the full scope of complex mixtures, ambient concentrations, and the prediction of realistic pollutant sources. For example, De Vito et al. have used arrays of low-cost sensors to detect and separate tertiary mixtures of compounds [10,11], and others have applied similar approaches to develop “e-noses” that can identify specific mixtures of a few compounds [12,13,14]. The primary difference between these studies and our own is that we have focused on identifying the likely source of the pollutants rather than attempting to identify individual compounds.

This study was also partially motivated by the problems understanding the outputs of low-cost sensors in the context of their increased noise and cross sensitivities relative to regulatory-grade instruments [15]. Our own field experience and recent papers from the low-cost air quality instrument community have consistently encountered difficulties understanding and interpreting the outputs of low-cost sensors, especially when communicating results to individuals who may not have a highly technical understanding of the instruments and/or atmospheric chemistry [16,17]. This study attempted to address some of these issues by providing an initial interpretation of the sensor outputs that can be produced in near real-time and are output in a way that a user could understand: the likelihood that a type of source is or is not affecting the measured air quality.

## 2. Materials and Methods

The methodology presented here was designed to accomplish two key goals. The first was to establish a system that is capable of detecting several compounds that are important air quality indicators. The second function of the system was to take those estimates and use them to identify likely source(s) of those pollutants. Specifically, this study aimed to develop a system that could identify pollutant sources that are often associated with VOC emissions. Note that the system here would not be more specific than a “class” of source like traffic, natural gas leaks, etc.

This can be understood to be similar to a typical measurement campaign intended to identify VOC sources. In these studies, researchers would sample for several compounds of interest using research-grade instruments. Using those measurements, the researchers would then attempt to determine the likely sources based on their knowledge of typical emission factors and co-emitted compounds.

To develop a preliminary low-cost source attribution tool, we assembled an array of diverse, low-cost sensors. We installed the array in an environmental test chamber and exposed it to simulated pollutant mixtures in a temperature- and humidity-controlled environment. We then adopted a two-step data analysis technique that would investigate the ability of the array to detect pollution and address source attribution. The first step of this process was a regression model that estimated the concentration of pollutant species using the sensor array signal as input. We investigated several regression models; these are discussed in the next section. The estimated concentrations were then used as the inputs to classification models that were trained to identify the source that was being simulated. Again, we investigated a diverse selection of classification models and compared the accuracy of source identification produced by each combination of regression and classification model.

### 2.1. Sensor Array Design

In order to detect a variety of gaseous compounds for inferring the presence of a local pollution source, it was necessary to develop an array of sensing elements that were sensitive to a varied mix of pollutants at relevant concentrations. This was accomplished by modifying the existing sensor platform, the Y-Pod [18,19]. The Y-Pod is an Arduino-based open-source platform that is typically configured to include nondispersive infrared (NDIR), photoionization detector (PID), metal oxide (MOx), and electrochemical types of low-cost sensors for gaseous compounds (see Figure 1). Two external boards with additional electrochemical and metal oxide sensors were added to the baseline Y-Pod configuration (see Figure 2). Two Y-Pods with this new configuration of multiple boards were tested with some of the sensors varying between the two Y-Pods. The sensors included on each board are listed in Table 1, and the data streams from both pods were combined and analyzed as if they were one large array.

It has been shown that changes in how sensors are operated and manufactured influence how those same sensors react to changes in gas concentrations and other environmental factors [20,21,22,23,24,25]. With this in mind, sensors were selected from multiple manufacturers and with different target gases to maximize the variance of sensor signal dependence to different gases between individual sensors. One important aspect underlying the differences between sensors is the presence of non-specific cross-sensitivities that affect sensor signals when exposed to non-target gases [5,7,26,27,28,29,30,31]. When attempting to quantify a single compound, these interferences are treated as an issue that must be corrected for, but it is proposed here that a high-dimensionality array of sensors could produce estimates of gases that are not the intended target of any single sensor. Some authors have investigated these cross-sensitivities and their usefulness when used in combination, but this field is still an area of active investigation [32].

Specific sensor models selected for use in this study (see Table 1) were determined after reviewing the literature to determine the sensitivity and usefulness of different models, as manufacturers rarely supply data from their sensors at concentrations that are relevant for ambient air monitoring. Because the low-cost sensor field is fast-moving, some of these sensors are no longer available for purchase. For example, the metal oxide CO_2_ sensor (TGS 4161) is no longer produced, and the MiCS-2710 has been replaced by the newer MiCS-2714. The variable name(s) associated with each of the selected sensors are listed in the table included in Appendix A.

#### 2.1.1. Metal Oxide (MOx) Sensors

Metal-oxide-based low-cost sensors are extremely common in the literature due to their small size and especially low cost (on the order of $10 USD/ea.). They have been used to measure a range of species that are important for air quality measurement and include NO_2_, O_3_, CO, VOCs, and CH_4_ [19,20,26,33,34,35,36]. In the default configuration of the Y-Pod there are three “onboard” MOx sensors. Of these onboard sensors, two are designed for Figaro MOx sensors, and the heaters are connected to an op-amp circuit designed to keep the heater element at a constant resistance (i.e., temperature) by providing a varied electrical current. The third MOx pad is designed for an e2v sensor and drives the heater with a constant voltage. All three onboard MOx sensors’ sensing elements are measured using a voltage divider and analog-to-digital converter (ADC) chips. Additional MOx sensors were added to each Y-Pod board by installing them in voltage dividers with additional ADCs that communicated with the main board via the I^2^C protocol (see Figure 2).

#### 2.1.2. Electrochemical Sensors

Electrochemical sensors are another popular low-cost sensor technology and have been used widely for the study of pollutants at ambient concentrations [5,7,26,27,30,37,38,39,40]. For the purposes of this study, all electrochemical sensors were those manufactured by Alphasense and consisted of models designed to target NO, NO_2_, CO, O_3_, and H_2_S with some sensor models duplicated. Alphasense electrochemical sensors were selected because of both the broad base of study that has been conducted with them in similar studies [5,26,30,38,40] and also because of the significant (claimed) variability in cross-sensitivities between the models selected for study here [29]. As shown in Figure 2, the electrochemical sensors are all installed on boards containing a total of four sensors, each of which is in a potentiostatic circuit as recommended by their manufacturer [41] and the induced current is measured using an op-amp circuit and ADC as also recommended by the manufacturer [41].

#### 2.1.3. Other Sensor Types

Nondispersive infrared (NDIR) and photoionization detection (PID) are two technologies that have been used in the literature for the detection of CO_2_ and total VOCs [7,36,39]. An NDIR sensor manufactured by ELT Sensor was included and was designed to be sensitive to CO_2_. This sensor communicates estimated CO_2_ concentration directly to the main Arduino board using the I2C protocol. The included PID was manufactured by Baseline Mocon and is sensitive to total VOCs and outputs a proportional analog signal that varies from 0–2.5 volts as the concentration ranges from 0–20 ppm, respectively. The Y-Pod also includes commercial sensors for the measurement of ambient temperature, relative humidity, and barometric pressure.

### 2.2. Environmental Chamber Testing

For this study, it was determined that testing in a controlled environmental chamber would allow us to best characterize the array and to determine the feasibility of its usefulness when exposed to complex gaseous mixtures in the field. The test chamber used for this study was Teflon-coated aluminum with a glass lid with a total volume of approximately 1500 in^3^ (24.6 L). In order to reduce the total volume, two glass blocks were added to the chamber, reducing the unoccupied volume to 1320 in^3^ (21.6 L). A small (40 mm, 6.24 ft^3^/min [177 Lpm]) fan was also installed inside the chamber in order to promote mixing throughout.

Gas mixtures were varied in both total concentration and composition to attempt to capture the variation in emissions within different classes of pollution sources. Both temperature and humidity were adjusted independently to capture variability in sensor responses to both environmental parameters. The test plan was conducted intermittently over three months to attempt to capture temporal drift, a factor known to affect many types of low-cost sensors [42,43]. The full summary of concentration and other environmental parameter values produced during testing is available in Appendix A.

#### 2.2.1. Simulating Pollutant Sources

We selected classes of sources to be simulated that are relevant to the communities in which these sensors could be deployed. The example community selected was one with which the Hannigan Group has performed another air quality research project and is located near downtown Los Angeles, California [44]. This community was selected because of its proximity to a variety of sources that may contribute to local air quality issues. These sources include oil and gas products and heavy traffic on major interstate highways, both of which can contribute to local elevations in concentrations of BTEX (benzene, toluene, ethylbenzene, and xylene) compounds and other VOCs. Produced liquids from oil and gas extraction were simulated using gasoline headspace, which contains hydrocarbons that can vaporize from produced oil and condensates. Additionally, natural gas leakage and low-NO_x_ combustion events (like biomass burning in wildfires) were selected as other sources relevant to the community.

Sources were simulated by injecting up to three component gases along with dilution air into the environmental chamber in a flow-through configuration. The dilution air for all testing was bottled “Ultra Zero” air, which is a synthetic blend of oxygen and nitrogen and contains less than 0.1 ppm of total hydrocarbons [45]. Flow control was accomplished using mass flow controllers (MFCs) that ranged from 0–20,000 SCCM for the dilution air and 0–20 SCCM for the component gases. Each MFC was calibrated in situ using bubble flow meters as reference flow measurements, and a linear calibration was applied to each MFC prior to calculating in-chamber concentrations. Reference gas concentrations were calculated as a flow-weighted average as shown in Equation (1), where *F_i,t_* is the concentration of the compound at the inlet of the chamber, *F_i,b_* is the concentration of the pollutant in the supply bottle, *Q_i,b_* is the flow rate of the gas from that bottle, and *Q_m_* is the flow rate as reported by each MFC, including the dilution air.

(1)$$F_{i,t}=\frac{\sum_{0}^{j}F_{i,b}*Q_{i,b}}{\sum_{0}^{n}Q_{m}},$$

Our test chamber did not include instrumentation to measure concentrations within the test chamber, so we assumed that the concentration of gas flowing into the chamber was equal to the concentration within the chamber. In order to make this a reasonable assumption, we determined the time between the inlet conditions changing and the concentrations stabilizing within the chamber. This stabilization time period was calculated as four times the time constant for the chamber, a value that was determined to be roughly 600 s at 8.5 L/min, the nominal flow rate used for testing. The lack of online reference measurements also necessitated the assumption that component gases not plumbed to the gas injection system for a given test were not present, i.e., that their concentration was exactly zero. This was a limitation of the system, but the use of “Ultra Zero” air for dilution made this a more reasonable assumption than if ambient air were used.

The component gases are listed in Table 2 for each simulated source. For the testing of each source, the total concentration of the mixture and the relative concentrations of each component gas were varied to attempt to better simulate variation within source classes while remaining representative of the source. Figure 3 illustrates the variation within and between sources with respect to the concentration of component gases. For example, the level of nonmethane hydrocarbons (NMHCs) within the simulated natural gas was varied from 0%, representing an extremely dry gas, to just over 20%, representing a rich gas that would be more representative of midstream or associated natural gas. This variation was also intended to try to remove some of the correlations between component gases to allow better understanding of which sensors were specific to which gas. The dilution ratio and ratios of component gases were limited by the flow ranges of the MFCs and the concentrations of the gas standards used to supply the component gases. For a full matrix of conditions for each test point, please refer to the Appendix A.

The two simulated combustion sources did not include VOC species, despite the associated VOC production motivating their inclusion in this testing. This decision was made for two reasons. First, the typical emission rates of primary compounds like CO, CO_2_, and NO_2_ from highway traffic are an order of magnitude higher than the emission rates for total gaseous VOCs on a mass basis, and even higher on a molar basis [46]. Secondarily, the NMHC emissions from combustion are dominated by highly reactive oxygenated volatile organic compounds (OVOCs) as well as semi-volatile organic compounds (SVOCs) [47], both of which are impractical to obtain as reference gas cylinders because of their propensity to react and/or condense. Because of these factors, it was decided to attempt to detect the primary pollutants (CO, CO_2_, and NO_2_) as indicators for the two types of combustion sources. NO_x_ was not included with “low-temperature combustion” both because NO_x_ emissions are dominated by diesel combustion [48] and because it has been shown to quickly react with other emitted species into PAN (peroxyacetyl nitrate) and particulates [49]. If a significant influence of mobile sources or biomass burning was detected, one could use typical emission rates to estimate VOC exposures and/or could bring a more precise and targeted instrument into the community to make regulatory or other important decisions.

#### 2.2.2. Simulating Other Environmental Parameters

In addition to simulating different mixtures of gases, temperature and humidity were also varied to attempt to capture environmental effects that are known to affect low-cost sensing devices [3,50,51]. Temperature control was accomplished using a resistive heater, and humidity control was accomplished by bubbling the dry dilution air through a water bath. Typical values ranged from 25–45 °C and 40–80%, respectively. Both temperature and humidity were recorded inside the chamber throughout testing, and a fan was installed inside the chamber to ensure that the temperature and humidity were uniform throughout the chamber.

### 2.3. Computational Methods

After subjecting the sensor arrays to a range of mixtures and environmental conditions, it was possible to develop training algorithms to quantify the component gases and predict the presence of each simulated source. This analysis was conducted using a custom script written in MATLAB. To begin this process, the data from both the sensor arrays and the chamber controls were collated and prepared for analysis. Because of the high dimensionality and nonlinearity of the sensor dataset, a two-step approach was adopted and is illustrated in Figure 4. In this approach, a regression model is first fitted to predict the concentration data for each gaseous compound or class of compounds, e.g., carbon monoxide or NMHC. Please note that in this paper, “predicting” a value does not indicate the estimation of a value in the future but rather is more generally used to describe the application of a trained model to use a vector of feature values to generate an estimate of, in this case, the concentration of a given compound. The application of this regression model created a pre-processing step that reduced the dimensionality of the dataset in a way that created new features that were useful on their own. The estimated concentrations of each compound were then used as features for a classification model that attempted to predict the class of source that was being simulated, e.g., “Natural Gas” or “Mobile Sources”. The classification models for each source were trained to allow for the prediction of multiple simultaneous sources, although multiple simultaneous sources were not tested in this capacity due to limitations on the number of MFCs in our test chamber.

#### 2.3.1. Initial Data Preprocessing

Both datasets were down-sampled to their 1-min median values to remove noise and transient changes in conditions that could not be assumed to be uniform throughout the chamber. During preprocessing, other data cleaning steps included the removal of 60 min after the pods were turned on (warm-up times). The temperature and relatively humidity values were also converted to Kelvin and absolute humidity, respectively, and the data from the stabilization period described earlier were removed. This ensured that enough time was allowed for the chamber to reach its steady state such that it was appropriate to assume that the inlet mixing ratio was representative of the actual mixing ratio within the chamber. Further feature selection like principal component analysis (PCA), partial least squares regression (PLSR), and other similar tools were not used to reduce the dimensionality of this data because many of the models included methods to reduce overfitting that could be associated with a large number of features. At the end of the preprocessing stage, there remained features representing signals from 28 sensor components using 41 features with 3280 instances.

The full set of data was split into ten sequential, equally sized portions for fitting and validation using k-fold validation. This was used to understand the ability of each model to perform on data on which it was not trained on. The sections of data used for training and validation were kept the same for both regression and classification models, so the source identification estimates that are labeled as “validation” used the concentration estimates produced when validating the regression models. This meant that the sources were identified using data that was not used for training either the regression models or classification models and represent a true set of validation data. More details for how the validation was conducted are included in the Appendix A of this report.

#### 2.3.2. Regression Models Applied

The first step in the analysis process involved regressing several “key” compounds (see Figure 3) that were selected as especially important for both detecting and differentiating the emissions from different classes of sources. Although the primary aim of this study was not to produce highly accurate estimates of gas concentrations, good estimates of gas concentration are useful both for the community in which the sensors are deployed and would likely be more useful for the classification models applied in the next step. The literature is yet not settled regarding which regression models are most useful when applied to low-cost atmospheric sensing, so a broad selection of models was applied here in order to better appreciate the sensitivity of the classification models to using inputs produced using different methods.

Several regression techniques that have been applied in the literature were explored here and assessed first on their ability to accurately model the concentrations of several different gases. These regression techniques included multiple linear regression (MLR), ridge regression (RR), random forests (RF), gaussian process regression (GPR), and neural networks (NN). Separate models were trained for each compound to avoid “memorizing” the artificial correlations that were present in this study but would not fully represent the diversity of mixtures that could be expected in a field deployment. Although a brief explanation of the models is included below, a thorough discussion of each model and its application is included in the Appendix A.

Multiple linear regression is one of the most popular forms of regression used to convert sensor signal values into calibrated concentration estimates and have been used in a range of applications with many low-cost sensor technologies [3,30,36,52,53,54]. Because of popularity and the relatively low computational costs, several forms of multiple linear regression were investigated. These models included two methods (stepwise linear regression and ridge regression) that attempt to reduce the effects of multicollinearity that can affect model performance in environments like atmospheric chemistry wherein many parameters are correlated with each other. The linear models included the following: “FullLM”, a multiple linear regression model that used every sensor input from the array; “SelectLM”, which included only data from sensors that were known to react to the current target gas, as determined by a combination of field experience and manufacturer recommendations; “StepLM” a linear model with terms selected by stepwise linear regression; and finally, “RidgeLM”, which was a multiple linear model with weights that were fitted using ridge regression.

In addition to the linear models discussed above, several nonlinear models were included with the hope that they could capture nonlinear relationships between the sensors including the cross-sensitivities that have been discussed earlier. These nonlinear models were random forest regression (“RandFor”), gaussian process regression (“GuassProc”), and neural networks (“NeurNet”). These three models are able to capture much more complex, nonlinear relationships between variables. It was suspected that these models that included more complexity might capture the interactions and cross-sensitivities that a simpler, linear model might miss. The specific nonlinear models selected apply significantly different methodologies to determine and model those relationships, which should also lead to different biases and errors. The reader is directed to the Appendix A for a detailed explanation of the tuning and application of these models.

#### 2.3.3. Classification Models Applied

After generating estimates of “key” compounds using each of the above regression approaches, classification algorithms were trained to identify the class of “source” that was being simulated, using the estimated concentrations at each timestep as features. Each was selected for a combination of its appearance in the literature and because it works significantly differently than the other methods included [10,55,56,57]. Those algorithms applied here included the following: “Logistic_class”, logistic regression with terms selected by stepwise regression; “SVMlin_class” and “SVMgaus_class”, support vector machines (SVM) with both a linear and Gaussian kernel, respectively; “RandFor_class”, a random forest of classifier trees; and “NeurNet_class”, a set of neural networks trained to predict the presence of each source. For all of the methods presented here, the classification model outputs were values ranging from 0 to 1, where 0 indicates high confidence in the absence of a source, and 1 indicates high confidence in the presence of a source. When comparing the results to the actual simulated source, a value greater than or equal to 0.5 indicated a prediction that the source was present and a value less than 0.5 indicated that it was absent.

Other classification methods that were not included here but are present in the literature include k-nearest neighbor classifiers [51,58], linear discriminant analysis [35,42,59], and other, more specialized techniques targeted at specific sensor technologies [21,58]. These were not included due largely to their similarities to other techniques applied here.

#### 2.3.4. Determining the Best Combination of Classification and Regression Models

The best combination of classification and regression model was determined by comparing F_1_ score as calculated using Equations (2)–(5), where *n* indicates the number of source classes and *i* indicates a specific source class. The notations tp, fp, tn, and fn represent the counts of true positives, false positives, true negatives, and false negatives, respectively. The F_1_ score is a popular metric when understanding the performance of pattern recognition algorithms because it provides a balanced indication of both the precision and recall [60]. Precision can be understood as the fraction of times that a model is correct when it predicts that a source is present, and recall is the fraction of times that a source is present and the model “catches it”. The F_1_ score uses the harmonic mean of the two values, which more heavily penalizes poor performance in either metric, a consideration that becomes important when the prevalence of classes is not balanced. The F_1_ scores that were used for model evaluation were those calculated on data that was left out of the training data for both regression and classification and are shown in Table 3 for each combination of models.
(2)$$F_{1,all}=\frac{1}{n}\times \;\underset{i=1:n}{\sum }F_{1,i}$$
(3)$$F_{1,i}=2\times \frac{precision_{i}\times recall_{i}}{precision_{i}+recall_{i}}$$
(4)$$precision_{i}=\;\frac{tp_{i}}{tp_{i}+fp_{i}}$$
(5)$$recall_{i}=\;\frac{tp_{i}}{tp_{i}+fn_{i}}$$

## 3. Results

### 3.1. Regression Results

After training each model, the accuracy was measured as the root mean squared error (RMSE). Other analysis metrics like the Pearson correlation coefficient, that use the variance of the reference values were difficult to interpret because some folds of validation data consisted entirely of concentration values that were exactly zero. These periods exist because some validation folds consisted of time periods when a compound was not being added to the chamber. The system used here does not have an online reference measurement of chamber concentrations, so it was assumed to be exactly zero. The RMSE values plotted in Figure 5 illustrate that the models perform acceptably but are generally not as accurate as has been achieved in the literature [5,27,37,52]. They also show that they are susceptible to overfitting to various degrees as shown by the differences between the results on training data (dark colored boxes) and testing data (light boxes). The relatively poor performance that is reflected in the RMSE values is also observed qualitatively when plotting the estimated concentrations versus reference values. For those plots, refer to Appendix A.

Interestingly, the variation between models when compared by their RMSE is not as great as one might expect. It also appears that the regularization as applied to the ridge regression did not totally address the problem of overtraining, as evidenced by the training RMSE values consistently underestimating the testing errors. This overtraining could indicate that the regularization didn’t balance out the increased variance caused by the addition of temperature and humidity interaction features introduced only for ridge regression. Neural nets and random forest regression typically performed best in our study, possibly explained by their ability to encode complexities like interactions, changes in sensitivity over time, and nonlinearities in signal response.

The relatively poor performance of the models, as compared to the literature, may be partially explained by design of the experiment. That is, this chamber study was designed to intentionally introduce a wide range of conditions containing known or suspected confounding gases at significantly elevated levels. This wide range of sources and concentrations would be unlikely to occur in a single location and it was, therefore, more challenging to accurately predict concentrations than in a typical environment. Reference values are also calculated based on MFC flow rates, which introduces the assumption that the chamber is fully mixed and has reached steady-state concentrations equal to that at the inlet. Both assumptions necessarily introduce additional uncertainties. Finally, the nature of chamber studies produces relatively discrete values for each reference instead of the continuous values that would be experienced in an ambient environment. This forces models to interpolate across a relatively wide range of conditions that are not seen in the training data; likely an important factor that may reduce the accuracy of the models.

### 3.2. Classification Results

At the end of the classification step, a set of values from 0 to 1 was output by each of the classification models for each simulated source. These values indicated the model’s prediction that a source was or was not present at each time step. In a field application, these values would be the indication of the likelihood that a source was affecting the measured air quality.

The results of each combination of classification and regression model were evaluated using empirical cumulative probability functions (CDFs) of the model outputs when a source was and was not present. An example of one of those plots is shown in Figure 6 for the best-performing set of models (“FullLM” and “RandFor_class”). By plotting a CDF of the output of the model, it was possible to see how often a model correctly predicted the presence of a source (score ≥ 0.5) or absence of a source (score <0.5) when the source was present (blue line) and was not present (red line). The magnitude of the score also indicated how confident the model was in its prediction, so values closer to 0 or 1 would be more useful to an end user because it would give a stronger indication that the source was actually absent or present. For example, if the classifier output a value close to 1 for traffic emissions, the user could be more confident that elevated concentrations were due to an influence of traffic pollution. In these plots, an ideal and omniscient model would roughly trace a box around the exterior; yielding a score close to 0 for all points where a source was not being simulated (red), and close to 1 for all points when a source was being simulated (blue). The CDF was plotted for classification models trained on both the reference values and the estimated values output by regression models. This allowed us to determine whether a model would or would not be able to separate the sources when given “perfect” concentration values. If not, then the model was unlikely to be useful when trained on less accurate estimates produced by regression models.

Another analysis tool that was used to select models were the confusion matrices that are shown in Figure 7 for each combination of regression and classification models. The confusion matrices plotted here illustrate the fraction of times each source was identified as present when each of the sources was being simulated. Confusion matrices are useful because they help us to understand what types of mistakes the model is making. For example, the top row of each confusion matrix shows the fraction of times that a model predicted that a source was present when no source was being actively simulated. A confusion matrix for an ideal model would be a matrix with the value of 1 along the diagonal and 0 elsewhere, indicating that it predicted the correct source every time. As mentioned before, the threshold was set as 0.5 for each model output, and each model was trained independently, so it was possible that multiple sources were predicted for a single timestep. This functionality was intentional because it would allow for the identification of multiple simultaneous sources, a situation that is likely to occur in reality.

Several of the models had problems with confusion between low- and high-temperature combustion that should have been easily differentiated by the presence of NO_x_. After reviewing the data, it appears that the NO_2_ electrochemical sensor failed approximately 2/3 of the way through the test matrix, likely hampering the ability of regression models to learn its importance in predicting NO_2_ concentrations. Interestingly, the regression model that used all of the sensor signals was the one that produced the best inputs to the classifier models. This may be because it did not discard other sensors that would have been possibly less useful for the regression of NO_2_ but that also did not fail near the end of the test.

## 4. Discussion

### 4.1. Classification Results

Using the criterion of maximum F_1_ score as defined in Equations (2)–(4), the best performance was accomplished by using the full multiple linear model (FullLM) to produce concentration estimates for a set of random forest models (RandFor_class) that estimated the sources. The performance of this pair is highlighted in Figure 8, which shows that the models were most effective at identifying gasoline vapors, and the second most successful classification was for natural gas emissions. The most common mistakes made by the model were confusion between low-temperature combustion and heavy exhaust. This makes intuitive sense because the only differentiating factor in this study was the presence of NO_2_ during heavy exhaust emissions, a compound that the regression models struggled to accurately predict.

### 4.2. Sensor Importance for Different Compounds

The sensors that were most influential when estimating the concentration of different gases was investigated by probing parameters from a few different regression models. This investigation will hopefully inform the design of future sensor arrays that are targeting some or all of the compounds discussed here. Specifically, we explored the sensor signals that were selected by stepwise regression, the unbiased importance measures for sensors from the random forest regression models, and the standardized ridge trace generated when fitting ridge regression models. By looking at which sensors were especially important to different models, it was possible to see which sensors were likely to be truly important for estimating the concentration of a gas.

#### 4.2.1. Terms Selected by Stepwise Regression

Figure 9 shows the sensors selected by stepwise linear regression for each compound of interest where the boxplots indicate the statistical importance of those sensors when they were selected. Many of the terms selected algorithmically match the sensors that we have used based on previous experience [52,62], although there were a few surprises that would suggest further investigation. For example, terms that were expected were the inclusion of the Figaro 2600 and 2602 for the detection of methane and nonmethane hydrocarbons (NMHC). The importance of the Baseline Mocon sensor for gasoline was also expected, as it is one of the few sensors that our research group has used to detect heavy hydrocarbons at sub ppm levels. Finally, the sole selection of the NDIR sensor for CO_2_ from NLT was expected as this technology is generally quite robust to cross-interferences and has been used successfully before in other studies. Some of the unexpected terms were the inclusion of the CO_2_ sensor for indication of CO and NO_2_, although this is likely because NO_2_ was only ever present when CO_2_ was also present, although not always at the same ratios, and the reverse was not true. CO is interesting in that there were some test points where only CO was present, although it is possible that there were not enough to counteract the more common correlation between the two compounds. It’s also interesting that the gasoline models consistently selected the Figaro 4161, which is an older metal oxide sensor that was marketed for the detection of CO_2_. Other sensors were selected for inclusion in only one or two of the cross-validation folds for NO_2_, CO, and NMHC, which indicates that their inclusion is not reliable and may be caused by a correlation unrelated to sensitivities.

#### 4.2.2. Random Forest Unbiased Importance Estimates

The random forests were trained such that they determined the variable to split at each node by using the interaction-curvature test as implemented by the MATLAB function “fitrtree”. This method allows the model to account for interactions between variables and produces unbiased estimates of the change in mean squared error (MSE) associated with the variable selected to split at each node. Because the decision of which variable to split on is evaluated for each tree at each node using different subsets of the full set of variables, importance estimates are generated for every variable and are determined on a wide range of subsets of the data. The ten most important variables—in this case sensor signals—for each compound are shown in Figure 10.

For NO_2_, we once again see that none of the sensors were selected to have a strong importance. This again is likely at least partly due to the failure of the NO_2_ sensor part way through testing. This is reflected in the high ranking of the elapsed time (“telapsed”) variable, as it would help separate the point of failure for that sensor. The H_2_S electrochemical sensor is again among the more important inputs, reinforcing the hypothesis that it is at least partially sensitive to NO_2_. Interestingly, the NO_2_ sensor does feature prominently in the CO importance plot. It is possible that the individual trees making up the random forest learned to screen the NO_2_ sensor when it had failed so as to be able to still use its common association with NO_2_, although it is not clear why that would be true when estimating CO and not NO_2_.

Both the methane and nonmethane hydrocarbon (NMHC) models are dominated by the Figaro 2600, which is the sensor that we have shown to be effective for similar applications [52,62]. It is possible that the NMHC are simply taking advantage of the partial correlation of NMHC with methane and not actually recording a sensitivity of the Figaro 2600 to other NMHC. This is reinforced by the lower ranking of the 2600 when regressing gasoline concentrations. In this model, the Baseline Mocon PID sensor and Figaro 2602 are selected, and their selection matches our experience with detecting heavier hydrocarbons. The elapsed time variable is relatively important here, which matches a drift in the Figaro 2602 that was noticed when reviewing the sensor values over the full test period.

The CO_2_ model is also dominated by the NDIR CO_2_ sensor from ELT, another instance where the models match expectations [19,52]. The gasoline model is also again dominated by the Baseline Mocon PID sensor and the Figaro 2602, which are two sensors that we have used to study those compounds in the past. The MICS-5121wp sensor by e2v is also relatively important, and it, too, is a sensor targeted at the detection of VOCs.

#### 4.2.3. Standardized Ridge Regression Coefficients

The use of ridge regression offered an opportunity to review which sensors contributed to models when using regularization to avoid overfitting. Because of this regularization, more initial terms could be included; in this case, interactions between each sensor and temperature, humidity, and both temperature and humidity were included in the input dataset. These are denoted by appending “:T”, “:H”, or “:T:H”, respectively, to the end of the sensor name. The methodology here initially used a range of regularization coefficients to produce a “ridge trace” plot that showed the standardized weight assigned to each variable as a function of that regularization coefficient (see Figure 11). By reviewing the ridge trace produced for each pollutant, it was possible to select a regularization coefficient that seemed to be an appropriate combination of reducing variance while avoiding underfitting. For this study, that value seemed to typically range between 1 and 10 for all pollutants, so a value of 5 was applied to all models for consistency and to avoid over-learning that hyperparameter. The ridge trace plots for each compound provide another insight into which sensors were important enough to the model that its importance to the model outweighed all but extreme values of regularization. This is indicated by weight values that are relatively high and stable, even at high levels of regularization (the right side of the plot).

In the NO_2_ ridge trace, most of the weights collapsed toward zero as regularization was increased, indicating that none were especially strongly correlated with NO_2_. That said, the H2S sensor was relatively resistant to regularization, indicating that it may have some sensitivity to NO_2_. The CO_2_ NDIR sensor term with temperature interaction term is also quite stable, although this may simply indicate that NO_2_ was sometimes co-emitted with CO_2_. It is important to again note that the NO_2_ sensor included in the study appears to have failed during the second half of testing, likely explaining its absence from this plot.

For CO, a strong dependence was seen with the NO_2_ sensor, although this may again be caused by the fact that NO_2_ was sometimes released at the same time as CO. The CO_2_ NDIR sensor is also relatively resistant to regularization, but this may be caused by the same phenomena. Interestingly, the electrochemical ozone sensor (with temperature interaction) was relatively strongly and negatively represented in the ridge traces. This may indicate that the catalytic surface on the ozone sensor is somewhat sensitive to carbon monoxide, although the electrochemical cell would be operating in “reverse” as ozone is an oxidizing agent and CO is reducing.

Nonmethane hydrocarbons (NMHC) seem to have been dominated by their inclusion as a component of some of the simulated natural gas emissions, as both Figaro 2600s were strongly weighted even with high regularization. Temperature itself was also selected, although it is not clear why this was selected rather than the terms that included temperature interactions.

Gasoline vapors produced a significantly different plot from NMHC, likely because they were not co-emitted with methane and were not dominated by the ethane and propane signals from the simulated natural gas emissions. These vapors were well captured by the Baseline Mocon PID sensor that is designed to quantify gaseous hydrocarbons. Interestingly, the parameter of just this sensor and the parameter with the interaction of this sensor with temperature and humidity were both selected strongly. This may indicate that the sensor has some sensitivity to environmental parameters as well as the mixture itself. The Figaro 2602 with humidity interaction was also selected strongly, which is an expected result as these vapors are the target of the 2602 and humidity is known to affect metal oxide sensors. One of the pressure sensors (BME) was also selected with almost as much weight as the Figaro 2600 sensor. After reviewing the data, we determined that this was due to a spurious correlation with pressure in the chamber, which was not controlled.

For CO_2_, several of the Figaro 2600 sensors were selected, although it is not clear whether that is representative of a true cross-sensitivity to CO_2_. It is, for example, possible that these are actually cross-sensitive to CO, which was co-emitted with CO_2_ in some of the testing. The CO_2_ NDIR sensor on one of the pods was selected, although not as distinctly as one might have predicted, based on the typical sensitivity of these sensors.

Finally, for methane, the Figaro 2600 sensor term was selected most strongly at higher levels of regularization. The Figaro 2600 sensor that was installed on the ancillary board and which operated at a slightly different operating temperature was also selected somewhat at a similar level to the Baseline Mocon PID sensor. Both BME pressure sensors were again selected, representing an incidental correlation with pressure.

## 5. Conclusions

In this study, we have shown the ability of current modeling and machine learning techniques to quantify several trace gases and identify likely sources for those trace gases using an array of low-cost sensors. The best combination of regression and classification methods for source identification was a linear regression using every sensor in the array followed by a random forest of classification trees. The F_1_ score as calculated on data that was held out during optimization and model fitting was 0.718, indicating a relatively strong combination of accuracy and precision. The success of these models was an unexpected result as our previous work suggested that a linear regression model with every sensor signal was likely to overfit and perform poorly on validation data. We also initially suspected that this model would fail to capture the interactions between sensors and environmental parameters like temperature and humidity. Indeed, as far as regression accuracy, many of the other regression models outperformed this model when predicting on validation data. That said, it is possible that the random forest classifier model was able to encode the interferences that would have been normally captured by the regression model while also taking advantage of signals that would have been removed or reduced by other regression models. If true, the omission of the regression step could be considered, although it would greatly increase the computational power required to train and apply the random forest classifier models.

In this study, we also reviewed the variable importance from three different regression models to investigate the value of sensors when quantifying different pollutants of interest. Although there were some issues caused by sensor failure, these may provide valuable insight to researchers designing the next generation of sensor platforms.

Going forward, it will be important to validate these methods in more realistic environments as chamber calibrations are known to be imperfect representations of field performance [18,63]. One example of a study that could provide valuable insights would be the placement of this sensor array in regions where different sources are likely to dominate, e.g., immediately next to a highway onramp and then next to a well during completion. By collecting data in these locations as well as e.g., rural sites, it would be possible to determine how well the identification of sources might function in situations where the qualitative levels are known. This has been demonstrated by Molino and others for the detection of single sources [26].

There are also other analysis techniques that could be used to possibly improve the results presented here. Specifically, combining the models into ensembles for both classification and regression, the ensembles could improve overall results by combining models that make “decisions” in different ways and are, therefore, less likely to make the same types of mistakes. The use of generative classification models could also improve the identification of sources. These models learn the characteristics of different classes instead of learning to separate one class from another. This type of classifier generally performs better on smaller training data sets. This would be important for work like that presented here because any economical calibration strategy will always represent a small training set relative to the range of environments in which a sensor could be deployed.

Finally, many sensors were not available for inclusion in this array due to limited availability. Including new sensors in the array could fill in gaps that were not covered by the relatively limited variety of sensors here. These new inputs could involve similar sensors operated in new ways, e.g., temperature-cycled MOx sensors that have been shown to improve sensitivity to VOCs and other gases [21,35,64]. It could also include sensors with wildly different operating principles like micro-GCs (gas chomatographs) [27] or with new targets like the low-cost PM sensors that could also be useful in the differentiation of sources.

## Figures and Tables

**Figure 1 sensors-19-03723-f001:**
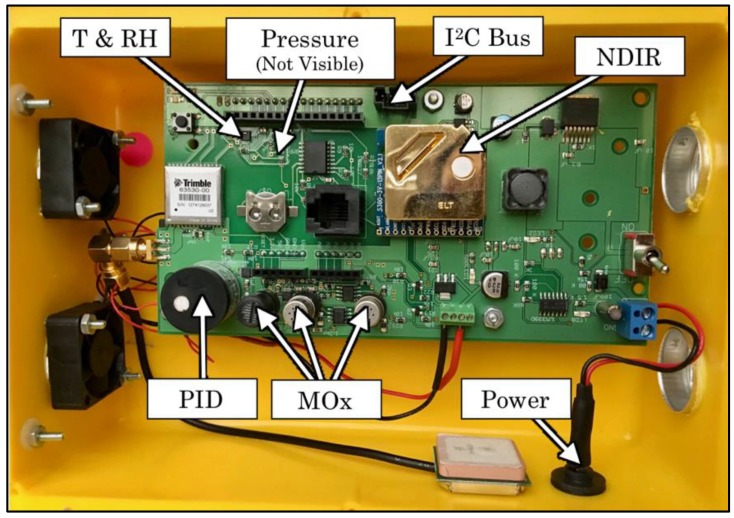
Image of a typical Y-Pod with important components labeled. The connector that is labeled “I^2^C bus” was used both to communicate with ancillary sensor boards and to provide power to those boards. The boards were not installed in the ventilated box shown in this photo but were instead placed directly into the test chamber.

**Figure 2 sensors-19-03723-f002:**
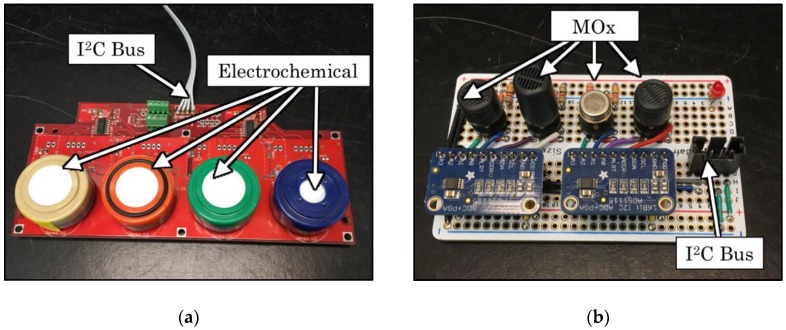
(**a**) Image of the additional boards that were used for electrochemical sensors and (**b**) the external board with additional metal oxide (MOx) sensors. The four-wire connections labeled “I^2^C Bus” contained both I^2^C communications and electrical power for the boards.

**Figure 3 sensors-19-03723-f003:**
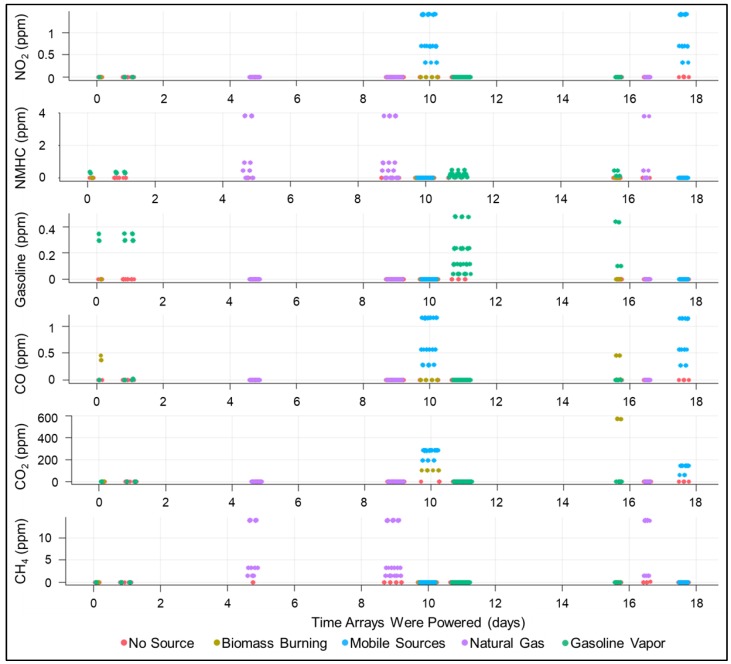
Timeseries of reference concentrations for each “key” component gas, colored by the simulated pollutant source. X-axis indicates elapsed time in days that the sensor arrays were powered. Gaps in the plots indicate times that they were powered but no test was being performed. During these times, the arrays were left in the chamber at room temperature and exposed to ambient indoor air.

**Figure 4 sensors-19-03723-f004:**

Information flow diagram illustrating how predictions of the sources of detected pollutants (in this case, natural gas) are generated indirectly from raw sensor data.

**Figure 5 sensors-19-03723-f005:**
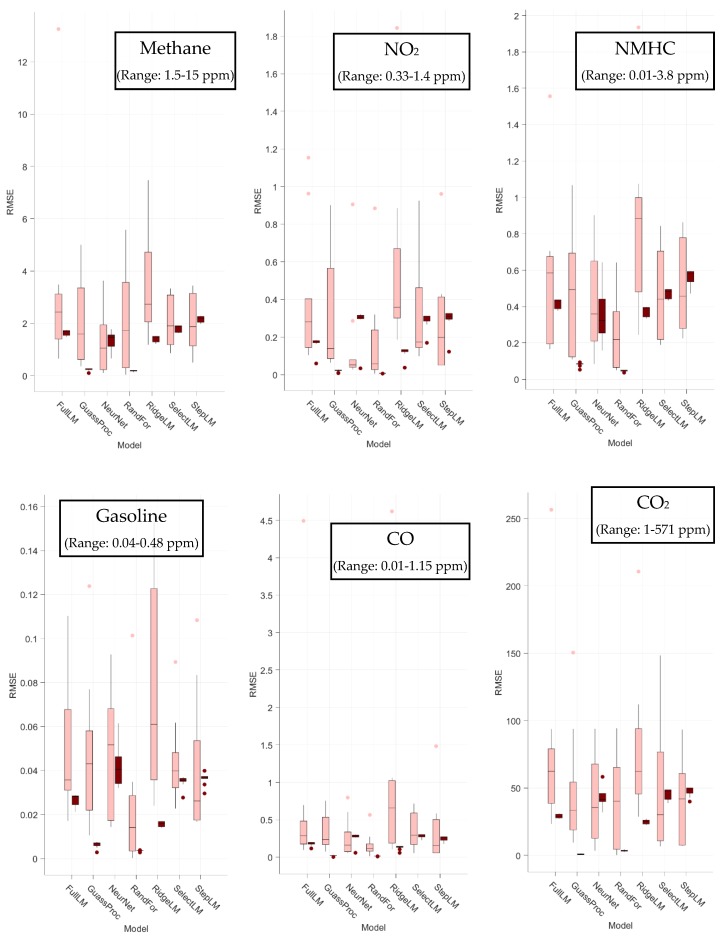
Boxplots of root mean squared error (RMSE) values (ppm) from each fold using each model and species. Bars indicate the 25th and 75th percentile values, whiskers extend to values within 1.5 times that range from the median, and dots represent points outside of that range as calculated using the GRAMM package implemented in MATLAB [61].

**Figure 6 sensors-19-03723-f006:**
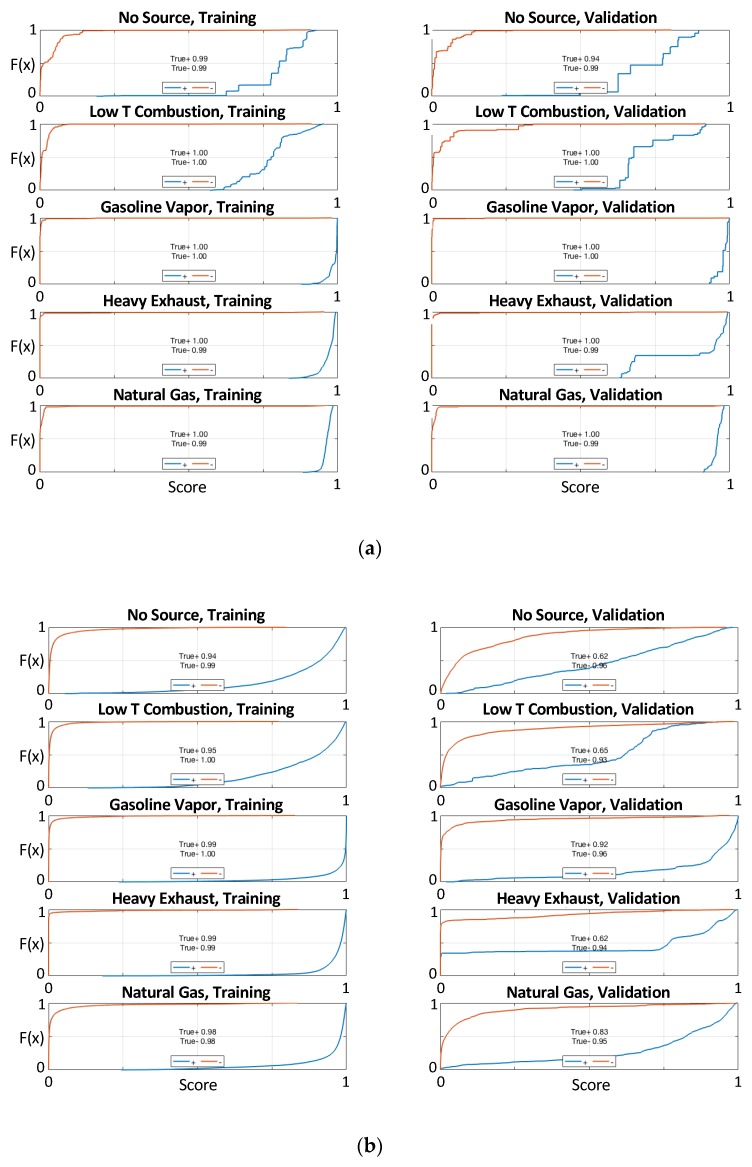
Example set of cumulative probability functions (CDFs) showing classification model performance on both calibration (train) and validation (test) data. The red lines trace the CDF for test cases where the source was not present (-), and blue lines illustrate the results when the source was present (+). (**a**) shows performance using the reference gas concentrations as features, and (**b**) shows performance for a combination of regression and classification models (in this case, FullLM and RandFor_class).

**Figure 7 sensors-19-03723-f007:**
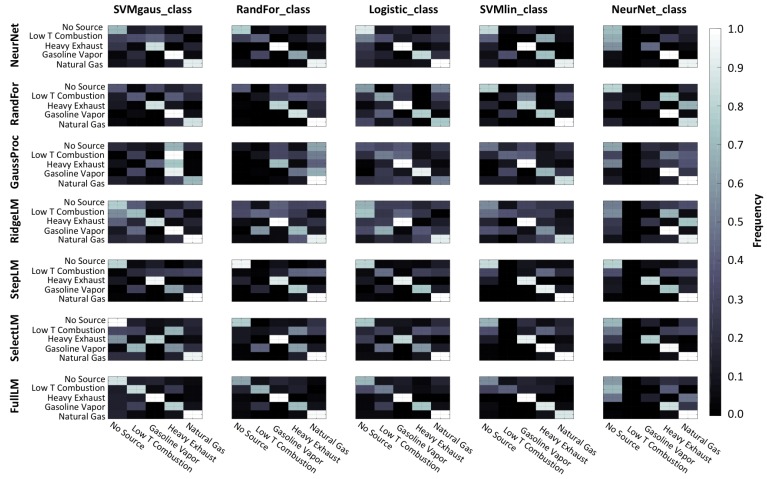
Confusion matrices showing the classification success on validation data. Each chart shows the results of a combination of classification (columns) and regression (rows) models. Within each confusion matrix, the reference class is shown on the “y” axis and predicted class is indicated on the “x” axis. Lighter colors indicate higher values, ranging from 0 to 1, which correspond to the fraction of times that a source was predicted when a source was being simulated.

**Figure 8 sensors-19-03723-f008:**
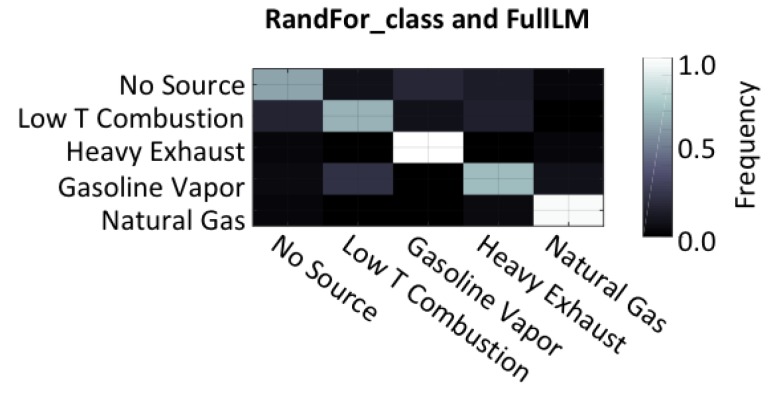
Confusion matrix for the “best” combination of models as judged by the F_1_ score as calculated using validation data. This combination involved regression using all linear models and then random forest classification trees.

**Figure 9 sensors-19-03723-f009:**
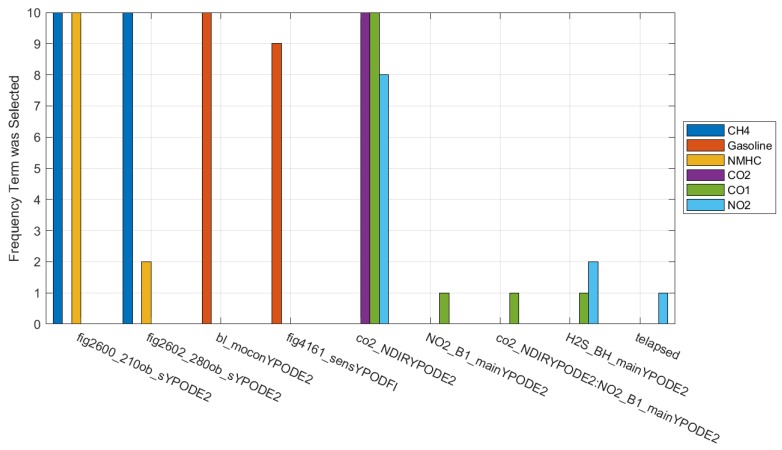
Frequency that each term was selected by stepwise regression for each pollutant from all cross-validation folds. When an interaction term was selected, it was named with “Sensor1:Sensor2” where “Sensor1” and Sensor2” are replaced with the sensor names.

**Figure 10 sensors-19-03723-f010:**
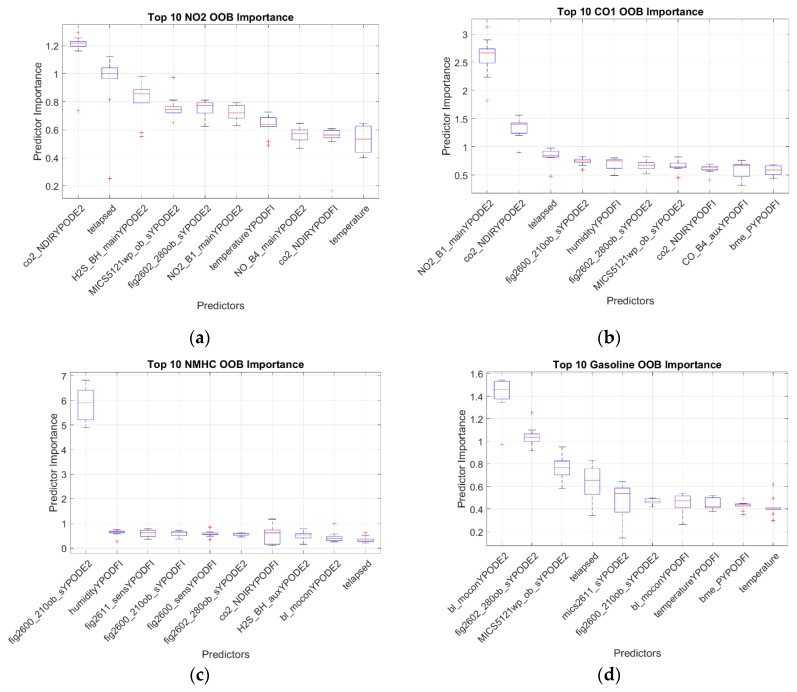
Box plots of the top 10 predictors sorted by out-of-bag importance estimates for each regressed pollutant. These pollutants were (**a**) NO_2_, (**b**) CO, (**c**) NMHC, (**d**) Gasoline, (**e**) CO_2_, and (**f**) CH_4_. Boxes indicate the variation in the predictor importance when models were trained on different cross-validation folds.

**Figure 11 sensors-19-03723-f011:**
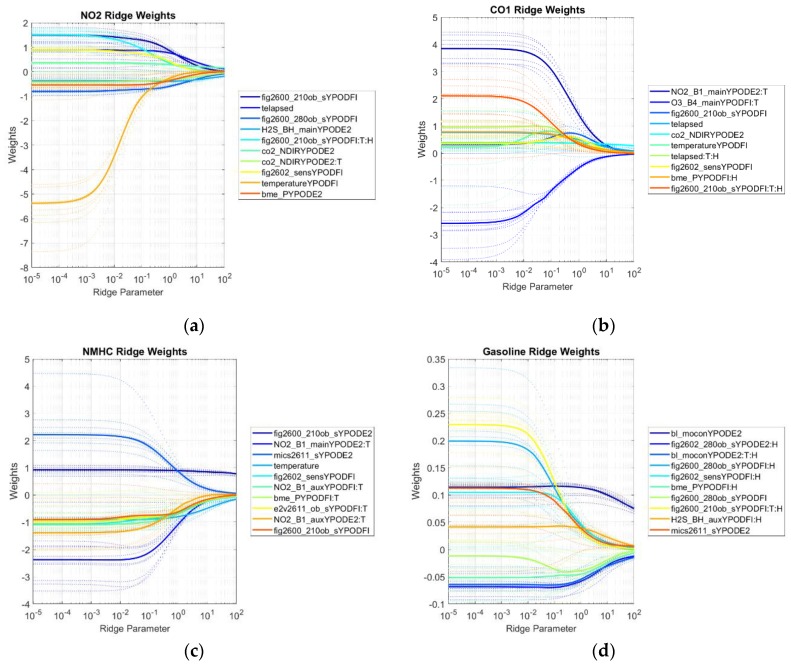
Summary of ridge traces of the top 10 most important parameters from the regression of each pollutant. These pollutants were (**a**) NO_2_, (**b**) CO, (**c**) NMHC, (**d**) Gasoline, (**e**) CO_2_, and (**f**) CH_4_. Weight magnitude should not be compared between compounds because they scale with the concentration values. Interactions with temperature and/or humidity are indicated by “:T”, “:H”, or “:T:H, respectively. Colors indicate the sensor and are sorted from largest to smallest importance at high regularization. Dashed lines illustrate the ridge trace for each cross-validation fold, and the solid lines illustrate the median value for all folds. The x-axis of each plot indicates the ridge parameter (regularization strength), and the y-axis indicates the unbiased parameter weight. Larger absolute values indicate a stronger importance of that sensor, especially at larger values of the ridge parameter.

**Table 1 sensors-19-03723-t001:** Details of the sensors that were selected for inclusion in the array.

Manufacturer	Model	Target Gas	Technology
Baseline Mocon ^1^	piD-TECH 0–20 ppm	Volatile Organic Compounds (VOCs)	Photoionization (PID)
ELT ^2^	S300	Carbon Dioxide (CO_2_)	Nondispersive Infrared (NDIR)
Alphasense ^3^	H2S-BH	Hydrogen Sulfide (H_2_S)	Electrochemical
	O3-B4	Ozone (O_3_)	Electrochemical
	NO2-B1	Nitrogen Dioxide (NO_2_)	Electrochemical
	CO-B4	Carbon Monoxide (CO)	Electrochemical
	NO-B4	Nitric Oxide (NO)	Electrochemical
Figaro ^4^	TGS 2602	“VOCs and odorous gases”	Metal Oxide
	TGS 2600	“Air Contaminants”	Metal Oxide
	TGS 2611	Methane (CH_4_)	Metal Oxide
	TGS 4161	Carbon Dioxide (CO_2_)	Metal Oxide
e2v ^5^	MiCS-5525	Carbon Monoxide (CO)	Metal Oxide
	MiCS-2611	Ozone (O_3_)	Metal Oxide
	MiCS-2710	Nitrogen Dioxide (NO_2_)	Metal Oxide
	MiCS-5121WP	CO/VOCs	Metal Oxide

^1^ MOCON Inc., Lyons, CO, USA. ^2^ ELT Sensor Corp. Bucheon-city, Korea. ^3^ Alphasense, Great Notley, Essex, UK. ^4^ Figaro USA Inc., Arlington Heights, IL, USA. ^5^ Teledyne e2v Ltd., Chelmsford, UK.

**Table 2 sensors-19-03723-t002:** Simulated source classes and the “key” species used to simulate them for the purpose of this chamber testing.

Source	Component Gases
Biomass Burning	CO, CO_2_
Mobile Sources	CO, CO_2_, NO_2_
Gasoline/Oil and Gas Condensates	Gasoline Vapor
Natural Gas Leaks	CH_4_, C_2_H_6_, C_3_H_8_

**Table 3 sensors-19-03723-t003:** F_1_ Scores calculated for different combinations of regression and classification models. Calculated using Equations (2)–(6) on data that was held out for validation during model training. Cells are colored by the indicated score.

		Classification Model				
		Logistic_class	NeurNet_class	RandFor_class	SVMgaus_class	SVMlin_class
**Regression Model**	FullLM	0.677	0.417	0.718	0.610	0.711
	GaussProc	0.416	0.401	0.394	0.288	0.535
	NeurNet	0.596	0.496	0.690	0.569	0.596
	RandFor	0.332	0.442	0.479	0.578	0.556
	RidgeLM	0.521	0.376	0.570	0.493	0.596
	SelectLM	0.600	0.531	0.545	0.493	0.596
	StepLM	0.695	0.502	0.619	0.614	0.616

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
