# Peer review of "Using A Low-Cost Sensor Array and Machine Learning Techniques to Detect Complex Pollutant Mixtures and Identify Likely Sources"

_sensors, 2019, doi:10.3390/s19173723_

Round 1
Reviewer 1 Report
The authors present a methodology that makes use of low-cost air quality sensors and computational intelligence methods to identify the likely source of pollutants in a two step approach; regression for data modelling and classification for source identification. This is an interesting paper contributing to state-of-the-art research in an international level, including sensor modelling and sensor interpretation. The authors are advised to considerably revise their document, by initially stating clearly the research questions that they address, the presenting materials and methods, followed by results and discussion. In terms of modelling, they should clearly describe the initial feature space, the feature selection and identification methods, and then the mode algorithms.
Comments in more detail follow:
· Line 18-19: Please re-phrase, it seems that there is something missing.
· Line 32, line 52: “secondary products like particulate”. Particles are not only secondary products but they can also be primary products of air emission processes (multiphase, multicomponent)
· Chapter 2.1.1: disadvantages of MOX sensors should be mentioned: low selectivity (due to cross-interferences) and questionable stability (they do not reproduce the same signal when exposed to the same concentrations, due to aging and chemical stress)
· Chapter 2.1.2: disadvantages of electrochemical sensors should be briefly mentioned: cross sensitivity (like CO to H2, O3 to NO2)
· Line 195, 196: square inches should be also mentioned in S.I (you can follow conversion guidelines from https://www.nist.gov/physical-measurement-laboratory/nist-guide-si-appendix-b)
· Fig 3: y-axis labels have a very small font, should be edited accordingly
· Line 283: the authors state that the regression model was fitted to predict values. Fitting means that a high-dimensional dataset of input features is used to estimate (mode) the parameter of interest (here concentration of specific pollutant(s)) for the same timestep as the one of the features. Prediction means that this modelling targets a time step after the one of the features.
· Chapter 2.3.1., line 295: the authors should provide with more information n the number and the nature of the initial parameters (features) included in their dataset. In addition, authors should comment on the use (or here the absence of use) of feature selection methods that would help them identify the most important features for effectively model the target parameter(s)
· Line 344-345: the assessment of regression models was based on the ability of the classification models to accurately identify the sources of the gases that the sensors were exposed to. The regression models shoud be accessed on the basis of model performance metrics.
· Lines 346-414: the authors use various regression models by employing each time a different set or subset of input data (features), thus making the results not directly comparable. In addition, the authors do not mention the software that they used for their calculation. The authors do not mention the “key” compounds that they were aiming at estimating with their regression models either. In addition, the absence of concentration values (concentrations equal to zero, as mentioned n lines 485-86), make the whole modelling approach questionable in terms of its design and implementation The authors should be more precise in describing the steps that they followed. In addition, the use of the RMSE as the only metric for model evaluation should be supported by the Pearson correlation coefficient between mode results and actual values. The relatively poor performance of the modes (line 507) may be attributed to what has been mentioned previously.
· classification: it is suggested that the authors make use only of the confusion matrix instead of a range of numerical values between zero and one to represent the level of “confidence” in identifying emission sources.
· Figure 7: very small and not easy to follow, can actually be omitted.
· Figure 8: very small and not easy to follow, can actually be omitted and replaces by a more clear (visually) confusion matrix example.
· The method for selecting the best classification and regression approach (presented for the first time in chapter 4.1, lines 573-587) should be deleted from the discussion section and move to the materials and methods section.
· Results on the terms selected by stepwise regression (chapter 4.2.1 should be presented before the presentation of model results). The same stands for 4.2.2
Reviewer 2 Report
The paper compares various machine learning techniques applied to low-cost air pollution sensors. The aim of the study is to identify the sources of pollution. Some comments:
1. It would be interesting to mention (perhaps in a footnote) that the MICS2614 is no longer manufactured (from 2017) and that the MICS 2710 has been replaced by the MICS 2714. Therefore, these sensors are old sensors that are no longer manufactured.
2. I found subsection 2.3 too long. Most of the material in this section can be found in machine learning books. For example, (i) subsection 2.3.2 "Model Validation" explains what a k-fold validation process is. I would mention that a 10-fold validation is used; (ii) subsection 2.3.3 "Estimating Concentrations with Regression Models" again is an explanation of the methods that can be found in books. Mention which ones are used, perhaps classified into linear and nonlinear. Idem with subsection 2.3.4 "Prediction of the presence of sources with classification models".
3. Why different regression models are used is not justified. For example, ridge regression and stepLR are used to avoid multicollinearity. An analysis of the correralion with the characteristics used in the matrix would probably help determine whether this multicollinearity exists and possibly an analysis with partial least squares or principal components could explain whether multicollinearity appears. Then apply regression after this analysis. Summarizing, instead of explaining the different models as mentioned in my point 2, It would be more interesting to justify the use of each model and what it brings to the study.
4. I would better explain the data set, e.g. size of the dataset and size of the training set and testing set.
5. I would explain which components of the array of sensors are used in each regression model. For example, for regressing NO2, which features were used ? As an example, many people for the alphasense electroquimical NO2-B1 sensor use NO2, O3, Temp and Relative Hum to regress NO2.
6. Fig 8: It is very difficult to observe the values in the boxes that are darker. Also, if the paper is printed, the values cannot be read. Perhaps the use of other colors would help to see the values. Idem in Figure 9.
